# Pathophysiological and Clinical Significance of *Crotalus durissus cascavella* Venom-Induced Pulmonary Impairment in a Murine Model

**DOI:** 10.3390/toxins15040282

**Published:** 2023-04-14

**Authors:** Ricardo G. Figueiredo, Marcos Lázaro da Silva Guerreiro, Elen Azevedo, Mateus Souza de Moura, Soraya Castro Trindade, José de Bessa, Ilka Biondi

**Affiliations:** 1Programa de Pós-Graduação em Saúde Coletiva, Universidade Estadual de Feira de Santana (UEFS), Feira de Santana 44036-900, Brazil; 2Departamento de Saúde, Universidade Estadual de Feira de Santana (UEFS), Feira de Santana 44036-900, Brazil; 3Laboratório de Animais Peçonhentos e Herpetologia (LAPH), Departamento de Biologia, Universidade Estadual de Feira de Santana (UEFS), Feira de Santana 44036-900, Brazil; 4Departamento de Ciências Biológicas, Universidade Estadual de Feira de Santana (UEFS), Feira de Santana 44036-900, Brazil

**Keywords:** inflammation, alveolar damage, vascular injury, *Crotalus durissus cascavella*

## Abstract

Crotalus venom has broad biological activity, including neurotoxic, myotoxic, hematologic, and cytotoxic compounds that induce severe systemic repercussions. We evaluated the pathophysiological and clinical significance of *Crotalus durissus cascavella* (Cdc) venom-induced pulmonary impairment in mice. We conducted a randomized experimental study, involving 72 animals intraperitoneally inoculated with saline solution in the control group (CG), as well as venom in the experimental group (EG). The animals were euthanized at predetermined intervals (1 h, 3 h, 6 h, 12 h, 24 h, and 48 h), and lung fragments were collected for H&E and Masson histological analysis. The CG did not present inflammatory alterations in pulmonary parenchyma. In the EG, interstitial and alveolar swelling, necrosis, septal losses followed by alveolar distensions, and areas of atelectasis in the pulmonary parenchyma were observed after three hours. The EG morphometric analysis presented pulmonary inflammatory infiltrates at all time intervals, being more significant at three and six (*p* = 0.035) and six and 12 h (*p* = 0.006). The necrosis zones were significant at intervals of one and 24 h (*p* = 0.001), one and 48 h (*p* = 0.001), and three and 48 h (*p* = 0.035). *Crotalus durissus cascavella* venom induces a diffuse, heterogeneous, and acute inflammatory injury in the pulmonary parenchyma, with potential clinical implications for respiratory mechanics and gas exchange. The early recognition and prompt treatment of this condition are essential to prevent further lung injury and to improve outcomes.

## 1. Introduction

Snakebite accidents represent a fundamental public health medical problem, especially for rural workers in developing countries [1,2]. In Brazil, between 2010 and 2020, 277,754 snake accident victims required medical care in an emergency room. Of these, 25,264 cases involved *Crotalus durissus*, with 240 deaths registered [3]. *Crotalus durissus* is a venomous snake species found throughout Central and South America. Rattlesnake venom induces broad biological activity, including alterations in the coagulation cascade, neurotoxic, myotoxic, and cytotoxic compounds that can induce severe systemic repercussions. The multisystem impairment provoked by Crotalus venom is clinically characterized by coagulopathies, respiratory and renal failure, hypertension, shock, and death [4,5].

The respiratory system plays a key role in immunological regulation, primarily protecting against pathogenic agents and inflammatory response modulation. *Crotalus durissus* venom is a potent neurotoxin capable of inducing neuromuscular block, swelling, and myonecrosis [6]. This effect can induce aggressive muscular alterations and respiratory system homeostasis. Although potentially severe, respiratory distress related to snake venom has not been widely reported in the scientific literature [7,8,9].

*Crotalus durissus* venom contains several enzymes and protein components that can induce biological effects, including myotoxins, phospholipase A2 (PLA2), L-amino acid oxidase (LAAO), phosphodiesterase (PDE), snake venom metalloproteases (SVMP), and serine proteases (SVSP) [10]. Crotoxin, constituted of PLA2 and crotapotin subunits, is one of the most studied toxins. PLA2 is an enzyme that hydrolyzes the phospholipids in cell membranes, leading to local and systemic myotoxicity, lymphatic injury, and edema [11]. In the respiratory system, PLA2-related neuromuscular paralysis of the bulbar and respiratory muscles may be life-threatening [12].

Snake venom metalloproteases, another group of toxins in *Crotalus durissus* venom, are enzymes that break down extracellular matrix proteins, such as collagen and elastin. These proteins provide structural support to the lungs and other organs. SVMP hydrolyzes structural components of the basement membrane, compromising the mechanical stability of pulmonary capillaries [12]. Therefore, *Crotalus durissus* venom can cause severe hemorrhagic disorders, inflammation, and pulmonary edema [13]. LAAO toxicity, primarily due to its enzymatic activity, producing hydrogen peroxide and other reactive oxygen species (ROS), is related to local tissue damage and pulmonary injury [14]. Venom-LAAO catalyzes the oxidation of different groups of amino acids, generating disturbances in blood coagulation and platelet aggregation. ROSs are highly reactive molecules that can damage endothelial components and increase the expression of inflammatory genes. Excessive release of ROS causes oxidative stress and may lead to severe acute lung injury (ALI) [15].

The treatment of lung injury induced by *Crotalus durissus* venom depends on the severity of the disease. In mild cases, supportive care, such as oxygen therapy and fluid management, may be sufficient. In severe cases, patients may require mechanical ventilation to support their breathing. In addition, antivenom therapy may be used to neutralize the venom toxins and prevent further damage to the lungs and other organs [12]. Antivenom therapy can neutralize venom toxins, preventing further damage to the lungs and other organs. The dose and duration of antivenom therapy depend on the severity of the envenomation and the response to treatment. Adverse reactions to antivenom, such as anaphylaxis, serum sickness, and renal failure, are rare, but they can occur and should be promptly recognized and treated [16].

Although non-invasive biopsy techniques for clinical support currently exist, the use of animal models continues to be an efficient form of obtaining information about tissue aggression and defense mechanisms. In this context, murine experimental models are included in studies that seek to correlate the interaction of snake venom with the immune system, as well as the repercussions of this action on the physiology and function of different organs [17,18].

Lung injury induced by *Crotalus durissus* snake venom is a serious health complication that can lead to ALI and acute respiratory distress syndrome (ARDS). This study aimed to investigate the pathophysiological and clinical significance of *Crotalus durissus cascavella* (Cdc) venom on the pulmonary parenchyma structure in a murine model. In addition, we proposed a comprehensive translational perspective for further research on early clinical diagnosis and treatment of acute lung injury in the first hours after venom exposure.

## 2. Results

### 2.1. Lung Histopathological Changes 

The pulmonary parenchyma structure of the control group remained normal (Figure 1A,B). One hour after inoculation with Cdc venom, the pulmonary parenchyma sections in the experimental group demonstrated both focal and diffuse inflammatory infiltrates, varying from mild to moderate, vascular congestion, hemorrhage, mild thickening, rupture of alveolar septa, necrosis areas, interstitial swelling, distension of the smooth muscles of the bronchioles, and bronchial constriction (Table 1; Figure 1C,D). 

After three hours, we observed intense polymorphonuclear inflammatory infiltrates in the peribronchial and perivascular regions and extensive breakdown of the pulmonary architecture associated with small areas of necrosis and hyperplasia in the bronchial-associated lymphoid tissue—(BALT). Moderate septal rupture and alveolar swelling, interstitial swelling, congested veins, ciliary alteration in the respiratory epithelium, hemorrhagic spots, and atelectasis were also observed (Table 1; Figure 1E,F).

At six hours, there was a decrease in peribronchial and perivascular inflammatory infiltrates. However, there was a significant increase in vascular congestion, necrosis, septal ruptures and loss, alveolar distension, and hemorrhagic spots (Table 1; Figure 2A,B). After 12 h, an increase in inflammatory activity and intense diffuse distributed areas of coagulopathy, vascular congestion, extensive necrosis, septal thickening, and rupture, as well as alveolar and bronchial smooth muscle distension, was observed (Table 1; Figure 2C,D).

At 24 h, a diffuse and peribronchial inflammatory process persisted with emphysematous areas, vasoconstriction, intense necrosis zones, septal wall thickening and rupture, bronchial epithelial hyperplasia, and bronchial and vascular muscle distension (Table 1; Figure 2E,F). At 48 h, the inflammatory infiltrates remained with areas of accentuated necrosis, vascular congestion, and the appearance of atelectasis (Table 1; Figure 3A,B).

Masson’s trichrome staining revealed matrix alterations with multifocal collagen deposits in the pulmonary parenchyma, which are more intense in the perivascular and bronchial regions (Figure 3C,D). Significant fibroblastic infiltration was noted after six hours. Morphometric analysis of the inflammatory kinetic in the pulmonary parenchyma of the control group did not show significant alterations. 

### 2.2. Morphometric Analysis of Lung Parenchyma

The morphometric analysis of lung parenchyma in the experimental groups identified polymorphonuclear diffuse and focal inflammatory infiltrates, specifically in the bronchial and perivascular topography. We noticed significant differences in inflammatory changes at three and six hours (*p* = 0.035) and six and 12 h (*p* = 0.006) (Figure 4A). The intensity of the inflammatory response was higher at 12 h. The morphometric quantification of pulmonary parenchyma necrosis areas indicated extensive tissue destruction, being statistically significant at the times of one and 24 h (*p* = 0.001), one and 48 h (*p* = 0.001), and three and 48 h (*p* = 0.035) (Figure 4B). The other times evaluated did not demonstrate any statistical significance. 

Alveolar distensions and ruptures in diverse areas of the pulmonary parenchyma were quantified and compared throughout the inflammatory process and from the moment of exposure to the venom, with significant differences being observed between all experimental groups and the control group. There was a significant difference between the control and experimental group in relation to the times one and 12 h (*p* = 0.027), one and 24 h (*p* = 0.001), and three and 24 h (*p* = 0.027) (Figure 4C). The quantification of collagen deposits at six, 12, 24, and 48 h did not demonstrate any statistical significance (Figure 4D).

## 3. Discussion

Acute lung injury (ALI) and acute respiratory distress syndrome (ARDS), induced by *Crotalus durissus* snake venom, are severe medical conditions that require prompt and appropriate clinical management. Treating these conditions involves supportive care, mechanical ventilation, and antivenom therapy. Few studies have examined pulmonary involvement in snakebite accidents [4,7,19], and the pathogenesis of respiratory failure by *Crotalus durissus cascavella* venom has not been fully elucidated. However, respiratory failure is frequently associated with ALI, mechanical distress, and respiratory muscle paralysis [20]. In snake accidents, involving *Crotalus* genus, this manifestation has been reported most frequently in the subspecies *Crotalus durissus* [6,21,22]. This study showed patterns of lung injury induced by *Crotalus durissus cascavella* venom with extensive necroinflammatory changes in the entire pulmonary parenchyma, including bronchi and bronchial damage, interstitial and alveolar swelling, matrix alterations, vascular congestion, coagulopathy with vascular rupture, and diffuse alveolar hemorrhage. These findings are compatible with the pathophysiologic subtract for ALI and the clinical expression of ARDS.

In crotalic accidents, the crotoxin can compromise neuromuscular function, blocking the transmission of neurotransmitters and impacting diaphragm muscle function, which may result, clinically, in muscle paralysis and respiratory failure [4,23,24]. Other toxins, such as lectins [25,26] and serine proteases [27], as well as hypotensive components [28,29], can act synergetically to potentialize diffuse coagulation in the small and larger veins, as observed in the present study. These coagulopathies can also increase thromboplastin [17,30] and induce progressive ischemic lesions, necrosis, and inflammatory changes. This phenomenon was observed and quantified at all time intervals. In addition, it is necessary to consider the extracellular effect of cytokines and signal leukocytes, consisting of platelet-activating inductor molecules and inflammatory mediators [31,32].

Early epithelial hyperplasia, as well as bronchial obliteration, atelectasis, vasoconstriction, and swelling, suggest significant impairment of respiratory mechanics and pulmonary structural damage induced by *Crotalus durissus* venom [6,23]. We observed septal ruptures, as well as alveolar distensions, capillary, and bronchial obliteration, at 24 and 48 h after venom exposure. Our data suggest a loss of integrity of the alveolar-capillary membrane, accentuating tissue acidity and necrosis, which may lead to respiratory failure, hemorrhage, and pulmonary infarction. These findings were confirmed by morphometric analysis.

Pulmonary emphysema is characterized by alveolar elastolysis, generally induced by extensive and significant alterations in the distal structure of the terminal bronchiole, which provokes dilation of the airways or destruction of the alveolar wall, resulting in loss of respiratory surface and a reduction in elastic tissue [33]. The emphysematous areas increase lung compliance, respiratory work, and anteroposterior thoracic diameter of the mice. In addition, the resultant ventilation–perfusion disturbance is directly related to the magnitude of the pulmonary inflammatory damage and subsequent fibroblast proliferation, increase in collagen synthesis, and critical alterations in the matrix of the emphysema pathogenesis [34,35]. Furthermore, the activity of inflammatory proteases secreted by leukocytes and neutrophilic elastase contributes to the destruction of elastic fibers and the inhibition of alpha1-antitrypsin (AAT). Under physiological conditions, type II pneumocytes synthesize AAT in sufficient concentrations to prevent alveolar elastolysis [36,37].

The clinical expression of histopathological lesions was detected early at three hours, including high respiratory rates and signs of respiratory distress. Progressive atelectasis after three hours is especially worrying, as ventilation–perfusion mismatch generates pulmonary shunt and impairment in respiratory mechanics [38,39].

Distal matrix alterations in the small airways and alveoli at six, 12, 24, and 48 h indicate inflammatory damage in several areas of the pulmonary parenchyma. These early matrix alterations suggest an elastance loss, usually seen in late and chronic respiratory failure [36], alerting the need for rapid clinical interventions in patients bitten by *Crotalus durissus* to avoid extensive and irreversible pulmonary damage. Similar findings were reported for bronchial muscle distension [21].

The progression of inflammatory infiltrates and bronchus-associated lymphoid tissue (BALT) hyperreactivity can be explained by systemic toxin injury. An exacerbated response of liberating inflammatory mediators and cytokines, principally polymorphonuclears, occurs in response to leukocyte recruitment and tissue damage [21,40,41].

The increase in pulmonary inflammatory injury, mostly related to neutrophilic proliferation, especially at one, three, 24, and 48 h, is partly explained by the release of macrophages chemotactic factors, epithelium disruption, and proinflammatory effects of the venom related to crotoxin component phospholipase [42,43]. However, inflammatory modulation with decreased neutrophilic infiltrates at six and 12 h might be related to transient early neutrophil senescence and apoptosis induced by extensive free radicals’ production due to intense phagocyte activity [44,45,46,47,48].

Supportive care is the cornerstone of the treatment of ALI induced by *Crotalus durissus* venom. Clinical interventions include oxygen therapy, fluid management, and correcting electrolyte disturbances and coagulopathy [49]. Supplemental oxygen therapy is essential to ensure adequate arterial oxygen content in hypoxemic patients. Our data showed early extensive necroinflammatory changes in lung parenchyma and clinical signs of respiratory distress, suggesting a potential risk of respiratory failure in the first hours after venom inoculation (Appendix A). Since we observed intense inflammatory infiltrates in the perivascular regions, vascular congestion, and interstitial swelling, fluid management might be critical to prevent fluid overload and pulmonary edema, which can worsen lung injury. 

We showed an early extensive breakdown of the pulmonary architecture associated with atelectasis. Lung ultrasound and computerized tomography of the thorax (Chest CT) are accurate diagnostic tools for assessing the severity of lung injury. Non-invasive ventilation (NIV) could reduce respiratory muscle workload and prevent atelectrauma in high-risk patients with ALI. Mechanical ventilation is often required in patients with severe ALI induced by *Crotalus durissus* venom. The goal of mechanical ventilation is to provide adequate oxygenation and ventilation while minimizing further lung injury. The use of positive end-expiratory pressure (PEEP) might prevent alveolar collapse and maintain adequate lung recruitment [48]. 

Other therapies may be used to manage ALI induced by *Crotalus durissus* venom. The morphometric analysis of lung parenchyma identified extensive polymorphonuclear inflammatory infiltrates, specifically in the bronchial and perivascular topography. Corticosteroids have been shown to reduce inflammation and improve oxygenation in some patients with ARDS, which may improve airway resistance and ventilation. However, the use of these therapies is controversial, and their efficacy and safety in the management of ALI induced by *Crotalus durissus* venom are unclear [49]. Preclinical findings of the efficacy of varespladib, a secreted phospholipase A2 inhibitor, was found to effectively inhibit the neurotoxic, anticoagulant, and myotoxic effects of snake venom [50,51,52,53]. Additionally, it prevented any further decreases in neuromuscular block. An ongoing phase 2 clinical trial (BRAVO study) aims to evaluate the safety, tolerability, and efficacy of oral varespladib-methyl in snakebite envenoming [54]. In vitro and in vivo preclinical venom inhibition assays of metalloproteinase inhibiting drugs also showed encouraging results [55]. Marimastat, batimastat, and prinomastat have been shown to be highly potent in their ability to inhibit SVMPs [52,56].

In *Crotalus durissus* accidents, patients can develop respiratory distress due to repercussions on the structure of pulmonary parenchyma and gas exchange, with potentially irreversible sequelae [46]. Our findings demonstrate the urgent need for further investigations into the respiratory repercussions and clinical conduction of patients bitten by *Crotalus durissus cascavella*. Early clinical interventions during the first 6 h of hospitalization, guided by point-of-care biomarkers, are critical to improve patient outcomes in high-risk subgroups and avoid long-term pulmonary sequelae [7,57]. Antivenom should be administered as early as possible, ideally within the first few hours after the snakebite [12]. Respiratory muscle paralysis and pulmonary edema remain the primary causes of death, with current treatments being based on clinical and respiratory support [49]. Thus, experimental histopathological analyses can provide valuable insights for clinical trials in this field. 

## 4. Conclusions

*Crotalus durissus* venom induces extensive pathophysiological changes, resulting in impaired pulmonary mechanics, poor gas exchange, and respiratory distress. The EG morphometric analysis presented pulmonary inflammatory infiltrates at all time intervals, being more significant between three and twelve hours. The necrosis zones were significant at intervals of one and 24 h, one and 48 h, and three and 48 h.

The early recognition and prompt treatment of this condition are essential to prevent further lung injury and to improve outcomes. Clinicians should be aware of the risk of extensive alveolar and vascular damage, which potentially lead to hypoxemia in the first hours after venom exposure. Assessment of alveolar collapse and severity of venom-induced lung injury evaluated by lung imaging might be clinically helpful. Further studies are needed to better understand the pathophysiology of ALI induced by *Crotalus durissus* venom and to develop novel biomarkers in ophidian envenomation. 

## 5. Method and Materials

### 5.1. Biological Material

The venom of Cdc was obtained from twenty adult snakes of both sexes, which was kept at the Venomous Animals and Herpetology Laboratory of the Feira de Santana State University (LAPH-UEFS). Venom was extracted, vacuum dried, and grouped into a “pool” at −20 °C for later analysis. The Laboratory is homologated by Instituto Brasileiro do Meio Ambiente e dos Recursos Renováveis (IBAMA), with a Federal Registration Number 480922, and it is administered by Sistema Nacional do Patrimônio Genético e do Conhecimento Tradicional Associado (SisGen), with its protocol number being ABC319C. Venom protein concentration was determined using bovine serum albumin from Sigma-Aldrich (St. Louis, MO, USA) as the protein standard [58].

A total of 72 male Swiss mice, weighing between 18 and 22 g, were randomly allocated into experimental and control groups (four experimental groups, four control groups, seven mice per group) and kept at controlled temperatures (22° and 25 °C), lighting (12-h light/dark cycle), humidity, and airflow conditions. All mice have also received water and food ad libitum. The animals in the experimental group received a 1.0 µg dose of venom per animal via intraperitoneal injection (i.p), diluted in 500 µL sterile saline solution (NaCl 0.9%), as well as the control group sterile saline solution (NaCl 0.9%), via i.p. [59]. The animals were monitored during the study interval. We euthanized animals with an overdose of xylazine chloride 2% (Xilazin^®^, Syntec, Santana do Parnaíba, SP, 06513-010, Brazil) and ketamine chloride 10% (Vetanarcol^®^, Konig, Santana do Parnaíba, SP, 06500-000, Brazil) to collect tissue samples. Lung tissue was initially conditioned in formaldehyde 10% for a maximum of 48 h (Figure 5). The behavior of the animals and the clinical manifestations were monitored during the experimental period. The study was conducted in accordance with the ethical principles for animal experimentation approved by the Animal Ethics Committee of the Feira de Santana State University (006/2018).

### 5.2. Histopathological Analysis of Pulmonary Parenchyma

The tissue fragments were processed, diaphonized, and sliced into 4 μm thick sections onto slides, which were later stained in a solution of hematoxylin-eosin (HE) and Masson’s Trichrome Sigma-Aldrich (St. Louis, MO, USA). The cuts were examined, and the images were captured using an Olympus BX51 microscope (Tokyo, Japan), coupled with a digital camera (DP25), and digitalized on cellSens software. 

### 5.3. Morphometric Analysis of Inflammatory Infiltrates Necrosis Areas, Alveolar Distensions, and Septal Loss

We conducted the morphometric evaluation in five randomly selected microscopic fields on slides containing parenchymal sections of the mice from the control and experimental groups. We evaluated inflammatory cell counting by square millimeters using a 20× eyepiece and 20× objective lens in an area of 60 mm^2^. The same methodology was used to quantify the necrosis areas, collagen deposits, alveolar distension, and septic loss by measuring and calculating alveolar saccule distensions and areas in the septic region that presented compromised integrity. All data obtained were processed and analyzed using ImageJ (USA) software. The graphics were generated using GraphPad Prism 6.0 (GraphPad, San Diego, CA, USA).

### 5.4. Statistical Analysis

After verifying the absence of normal distribution using the Kolmogorov-Smirnov test, the Mann-Whitney test was used to compare the experimental and control groups, and the Friedman test, followed by the Wilcoxon test (with Bonferroni correction), were used to compare time intervals. Values of *p* < 0.05 were considered statistically significant. Statistical analyses were conducted using GraphPad Prism, version 6.00 (GraphPad, San Diego, CA, USA).

## Figures and Tables

**Figure 1 toxins-15-00282-f001:**
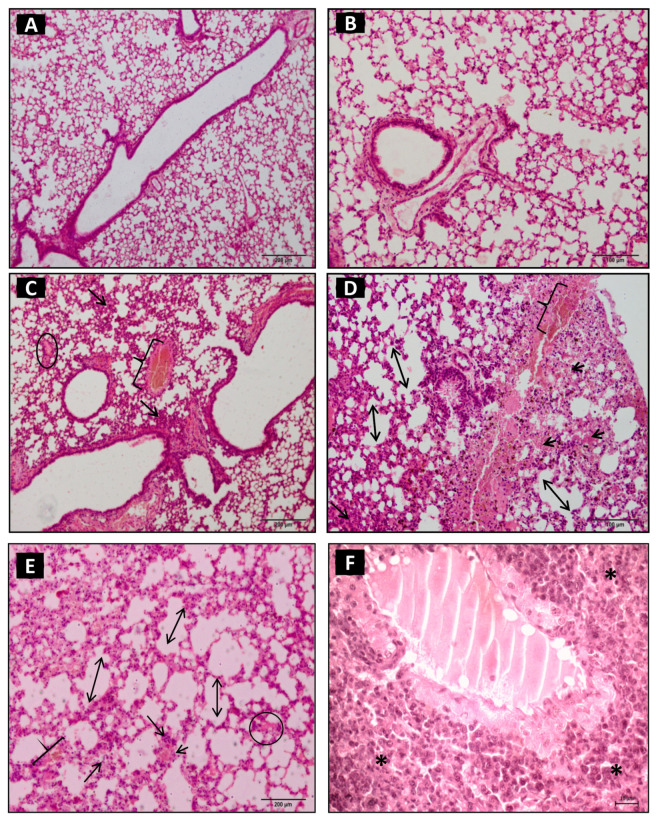
Photomicrography of pulmonary parenchyma of control group (**A**,**B**): inoculated with 500 μL of sterile saline solution (NaCl 0.9%) (i.p.) presented preserved pulmonary architecture without physiopathological alterations. The animals of experimental group (EG—(**C**–**F**)) inoculated with 1.0 µg venom/animal of *Crotalus durissus cascavella* diluted in 500 µL sterile saline solution (NaCl 0.9%) (i.p.). (**C**,**D**): one hour after inoculation, the pulmonary parenchyma presented perivascular inflammatory infiltrates (arrow), hemorrhagic spots (arrow head), septal loss, and alveolar distension (two-headed arrow), congested veins (keys), and swelling (circle). (**E**,**F**): after three hours, the presence of intense inflammatory infiltrates (arrow) with perivascular polymorphonuclear cells, septal loss (two-headed arrow), congested veins (key), interstitial swelling (circle), and atelectasis (asterisk) was observed. (HE, 20× objective).

**Figure 2 toxins-15-00282-f002:**
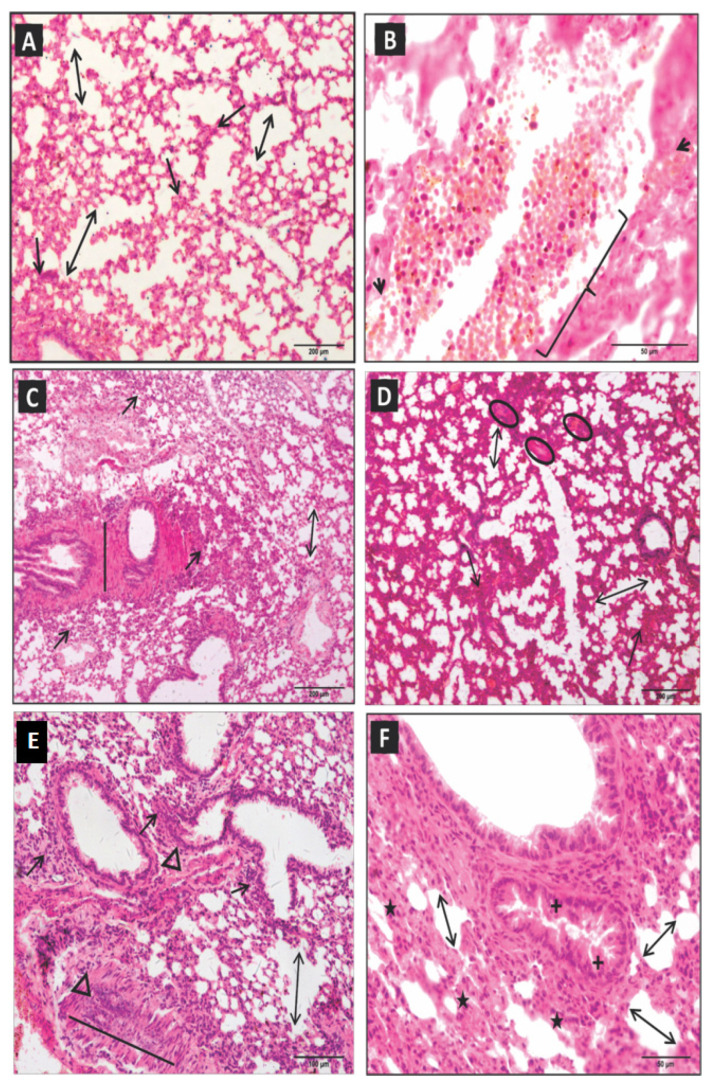
Photomicrography of pulmonary parenchyma of experimental group inoculated with 1.0 µg venom/animal (i.p.), diluted in 500 µL sterile saline solution (NaCl 0.9%). (**A**,**B**): After six hours, vascular congestion (key), septal loss with alveolar distension (two-headed arrow), and the presence of inflammatory infiltrates (arrow) and hemorrhagic spots (arrowhead) occurred. (**C**,**D**): After 12 h, the pulmonary parenchyma presented diffuse inflammatory infiltrates (arrow), septal loss (two-headed arrow), areas of coagulopathy (ellipse), and bronchial smooth muscle distension (straight line). (**E**,**F**): At 24 h, diffuse inflammatory and moderate peribronchiolar infiltrates (arrows), emphysema areas (stars), septal loss (two-headed arrow), vasoconstriction (triangle), vascular muscle distension (straight line), and bronchial epithelium hyperplasia (cross) were observed. (HE, 20× objective).

**Figure 3 toxins-15-00282-f003:**
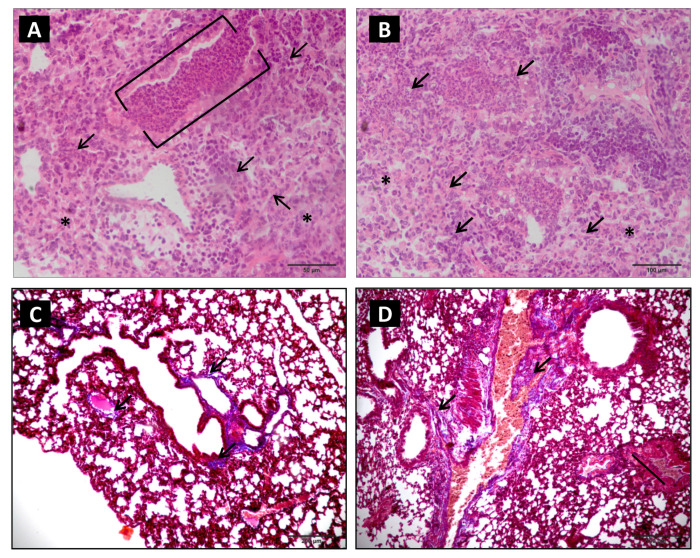
Photomicrography of pulmonary parenchyma of experimental group inoculated with 1.0 µg venom/animal (i.p.), diluted in 500 µL sterile saline solution (NaCl 0.9%). (**A**,**B**): After 48 h, extensive areas of parenchymal destruction, diffuse infiltrates, intense perivascular and bronchial spots (arrows), bronchial obliteration by inflammatory infiltrate (bracket), and atelectasis (asterisk) were observed, (HE, 20× objective). (**C**,**D**): At 12 and 24 h, respectively, a mild alteration in collagen deposits in the peribronchial and perivascular pulmonary parenchyma (arrows) was observed (Masson’s trichrome staining).

**Figure 4 toxins-15-00282-f004:**
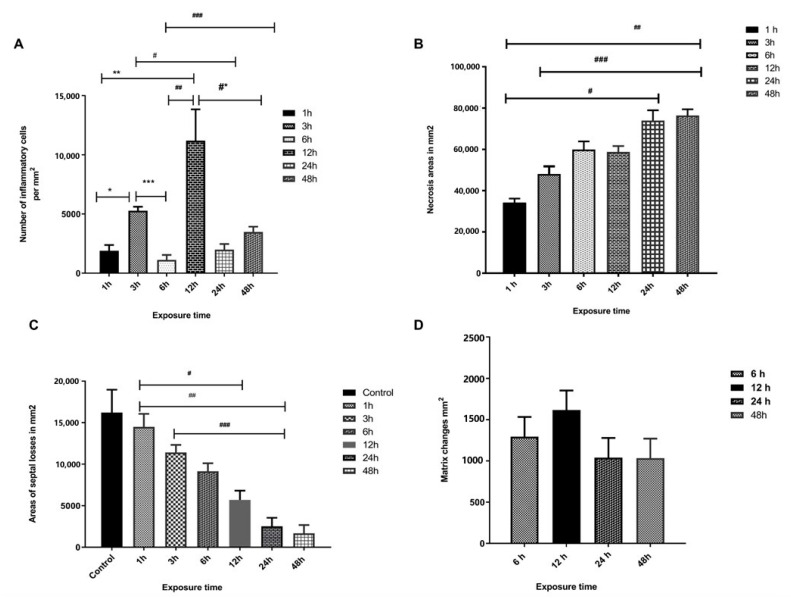
Morphometric analysis of the pulmonary parenchyma of experimental group inoculated with 1.0 µg venom/animal (i.p.), as well as diluted in 500 µL sterile saline solution (NaCl 0.9%) (**A**): Frequency of inflammatory infiltrates in the perivascular regions, predominantly polymorphonuclears, one and three (*p*= 0.0079 *), one and 12 h (*p* = 0.0079 **), three and 24 h (*p* = 0.0079 #), three and six hours (*p* = 0.035 ***), six and 12 h (*p* = 0.006 ^##^), six and 48 hours (*p* = 0.0159 ^###^), at 12 and 48 h (*p* = 0.0159 *#). (**B**): Areas of the parenchymal necrosis with significance difference at one hour and 24 h (*p* = 0.001 ^#^), one hour and 48 h (*p* = 0.001 ^##^), and three and 48 h (*p* = 0.035 ^###^). (**C**): Alveolar distensions and ruptures in diverse areas of the pulmonary parenchyma during the evolution of the inflammatory process with significant differences between the control group (CG) and experimental group (EG) in relation to the times one and 12 h (*p* = 0.027 ^#^), one and 24 h (*p* = 0.001 ^##^), and three and 24 h (*p* = 0.027 ^###^). (**D**): The quantification of the collagen deposits was performed at six, 12, 24, and 48 h. However, there was no statistically significant difference at the times analyzed. Values of *p* ≤ 0.05 were considered statistically significant.

**Figure 5 toxins-15-00282-f005:**
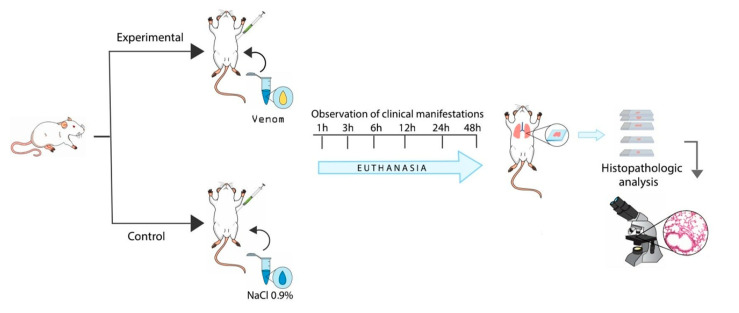
Experimental protocol.

**Table 1 toxins-15-00282-t001:** Semi-quantitative analysis of the action of *Crotalus durissus cascavella* venom on the pulmonary parenchyma at different time exposures in Swiss mice. Semi-quantitative representation of lesion intensity and inflammatory infiltrates (mild (+), moderate (++), and intense (+++)). Exposure time in the pulmonary parenchyma.

Exposure Time in the Lung Parenchyma	Inflammatory Infiltrates	Necrosis Areas	Septal Losses	Alveolar Distension
1 h	++	+	+	+
3 h	+++	+	++	+
6 h	+	+++	+++	+++
12 h	+++	+++	+++	+++
24 h	+++	+++	+++	+++
48 h	+++	+++	+++	+++

Semi-quantitative representation of the intensity of the lesions and the mild (+), moderate (++), and intense (+++) inflammatory infiltrates.

## Data Availability

All relevant data are within the manuscript and its Appendix A.

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
