# Peer review of "Pathophysiological and Clinical Significance of Crotalus durissus cascavella Venom-Induced Pulmonary Impairment in a Murine Model"

_toxins, 2023, doi:10.3390/toxins15040282_

Round 1

Reviewer 1 Report

The manuscript described the histopathological change and clinical significance of Crotalus durissus cascavella venom-induced pulmonary impairment in mice, and  analyzed the pulmonary inflammatory infiltrates and necrosis zones at time intervals. Although the venom showed manifested acute inflammatory injury in the pulmonary parenchyma, the experimental method and perspective used in the manuscript was relatively simple. The toxin and mechanism remains obscure, thus I believe the  manuscript in current state doesnt meet the publication standard of Toxins

Author Response

We are grateful to the reviewer for their insightful comments on our paper. We have been able to improve our manuscript quality along peer review process. Here is a point-by-point response to the reviewers' comments and concerns. I hope the manuscript in its current state will be suitable for publication in Toxins.

Reviewer 2 Report

Very nice work--there is not enough natural history of disease work in the field and this will add baseline data from which other studies or this laboratory can readily build. 

A few comments: 

1. Line 102--n=7/group? Do you mean n=6 (6x12=72)

2. Line 127 mm2 (should be superscript)

3. throughout: Italics for Genus species would be appropriate. 

General comment: 

The authors talk about limitations/controversy of existing therapies. There are several new approaches in the pipeline, including ongoing clinical trials that both lend themselves to direct investigation within this project (e.g. would varespladib or marimastat or prinomastat or combination of sPLA2/SVMP inhibitors reduce the pathological findings reported in this paper?). Varespladib is reported to inhibit C. d.t. venom sPLA2 in low nanomolar range. Marimastat/batimastat/prinomastat have all been reported as highly potent svMP inhibitors. Others have reported that the combination of varespladib and marimastat might be effective in some viper envenoming. 

To make this paper more forward looking, it would probably make sense to comment on what should happen next. 

Finally, I wonder if the investigators looked at or plan to look at other tissues? Renal tissues are mentioned twice as affected by this venom and are readily assessed. It could be mentioned here and then appear as another contributory manuscript to the field, for example.  

Great work. Thank you for the opportunity to review. 

Author Response

We are grateful to the reviewer for their insightful comments on our paper. We have been able to improve our manuscript quality during the peer review process. Here is a point-by-point response to the reviewers' comments and concerns. I hope the manuscript in current state will be suitable for publication in Toxins.

Reviewer 2

Line 102--n=7/group? Do you mean n=6 (6x12=72)

Response: We have revised the text in order to make our point clearer, as follows: “A total of 72 male Swiss mice, weighing between 18 and 22g, were randomly allocated into experimental and control groups (4 experimental groups; 4 control groups; seven mice per group)”

Line 127 mm2 (should be superscript)

Response: Revised accordingly.

throughout: Italics for Genus species would be appropriate. 

Response: Revised accordingly.

General comment: 

The authors talk about limitations/controversy of existing therapies. There are several new approaches in the pipeline, including ongoing clinical trials that both lend themselves to direct investigation within this project (e.g. would varespladib or marimastat or prinomastat or combination of sPLA2/SVMP inhibitors reduce the pathological findings reported in this paper?). Varespladib is reported to inhibit C. d.t. venom sPLA2 in low nanomolar range. Marimastat/batimastat/prinomastat have all been reported as highly potent svMP inhibitors. Others have reported that the combination of varespladib and marimastat might be effective in some viper envenoming. 

Response: We agree with the reviewer that more quantitative information about new approaches in snakebite envenoming would add relevance for the reader. We have revised the text, which we present below:

Page 9 Line 625: Preclinical findings of the efficacy of varespladib, a secreted phospholipase A2 inhibitor, was found to effectively inhibit the neurotoxic, anticoagulant, and myotoxic effects of snake venom [50-53]. Additionally, it prevented any further decreases in neuromuscular block. An ongoing phase 2 clinical trial (BRAVO study) aims to evaluate the safety, tolerability, and efficacy of oral varespladib-methyl in snakebite envenoming [54]. In vitro and in vivo preclinical venom inhibition assays of metalloproteinase inhibiting drugs also showed encouraging results [55]. Marimastat, batimastat, and prinomastat have been shown to be highly potent in their ability to inhibit SVMPs [52,56].

To make this paper more forward looking, it would probably make sense to comment on what should happen next. 

Response: Unpublished data from our group confirmed that the commercial Antivenom serum currently used in Brazil is ineffective against PLA2 of this venom. Considering that treatment with commercial serum is the only strategy adopted by clinical medicine in Brazil, we are expanding our investigations regarding the pathological lesions caused by isolated PLA2, including new strategies to inhibit the deleterious action of PLA2.

Finally, I wonder if the investigators looked at or plan to look at other tissues? Renal tissues are mentioned twice as affected by this venom and are readily assessed. It could be mentioned here and then appear as another contributory manuscript to the field, for example.  

Response: We have already completed the analysis to identify systemic effects in other target organs, including renal tissue. This manuscript is under peer review. We look forward to publishing it until June 2023.

Reviewer 3 Report

The reviewed article is interesting in terms of the results presented, but in this form it is not suitable for publication in Toxins. The following is a list of comments and questions that I ask the authors to consider as they revise the article:

General note to the entire article: please note that Latin names should be written in italics, both when the full name of the species appears and next to the generic name itself.

Line 56-57 "When metalloproteinases are activated by Crotalus durissus venom..." what do the authors mean? How does the venom activate metalloproteinases and which metalloproteinases are they referring to?

Line 64 "These include cytokines, chemokines, and ROS" please provide a citation confirming the presence of cytokines and reactive oxygen species in snake venoms. The citation provided does not exist.

Line 93 From how many specific individuals was the venom taken. How did they differ in age, sex, etc.?

Line 99. What method was used to measure the protein concentration in the venom, since the cited item (12) does not exist.

Table 1, Figure 5a Please elaborate on the observed changes in the discussion. How is the decrease in inflammatory infiltrates at 6h after venom application and the number of inflammatory cells at 6 and 24h explained. Also, please note whether the commentary in the discussion in lines 299-306 is sure to describe the correct time points.

Line 310 "Our data highlight the potential risk of early hypoxemic respiratory failure in the first hours after venom inoculation" Please indicate which specific results directly indicate this.

Excerpt "Antivenom therapy is the specific treatment for snakebite envenomation and is essential to neutralize the venom toxins and prevent further tissue damage. Antivenom should be administered as early as possible, ideally within the first few hours after the snakebite" is unrelated in content to the earlier paragraph and has no citation. Please reword this passage.

My biggest complaint is with the list of cited literature. Items 10 through 14 do not exist. That is, there are articles with similar titles, but they are not published in the journals indicated by the authors. In the case of item 10, there is an article with this title in this journal, but written by other authors and almost 20 years earlier. In addition, in item 13 there are two citations, both non-existent. This makes half of the introduction written without support in the scientific literature. Moreover in item 41, the names of the authors are mixed up.

Author Response

We are grateful to the reviewer for their insightful comments on our paper. We have been able to improve our manuscript quality during the peer review process. Here is a point-by-point response to the reviewers' comments and concerns. I hope the manuscript in current state will be suitable for publication in Toxins.

General note to the entire article: please note that Latin names should be written in italics, both when the full name of the species appears and next to the generic name itself.

Response: Revised accordingly.

Line 56-57 "When metalloproteinases are activated by Crotalus durissus venom..." what do the authors mean? How does the venom activate metalloproteinases and which metalloproteinases are they referring to? Line 64 "These include cytokines, chemokines, and ROS" please provide a citation confirming the presence of cytokines and reactive oxygen species in snake venoms. The citation provided does not exist.

Response: We have revised the text in order to make the role of metalloproteases in snake venom clearer. We inserted a new sentence explaining reactive oxygen species (ROS) role in lung injury in the follow paragraph:

Page 2 Line 59:  “Snake venom metalloproteases, another group of toxins in Crotalus durissus venom, are enzymes that break down extracellular matrix proteins, such as collagen and elastin. These proteins provide structural support to the lungs and other organs. SVMP hydrolyze structural components of the basement membrane, compromising the mechanical stability of pulmonary capillaries [12]. Therefore, Crotalus durissus venom can cause severe hemorrhagic disorders, inflammation, and pulmonary edema [13]. LAAO toxicity, primarily due to its enzymatic activity producing hydrogen peroxide and other reactive oxygen species (ROS), is related to local tissue damage and pulmonary injury [14]. Venom-LAAO catalyzes the oxidation of different groups of amino acids generating disturbances in blood coagulation and platelet aggregation. ROS are highly reactive molecules that can damage endothelial components and increase the expression of inflammatory genes. Excessive release of ROS causes oxidative stress and may lead to severe acute lung injury (ALI) [15].”

  1. Gutiérrez JM, Calvete JJ, Habib AG, Harrison RA, Williams DJ, Warrell DA. Snakebite envenoming. Nat Rev Dis Primers. 2017 Sep 14;3(1):1–21.
  2. Gutiérrez JM, Rucavado A, Escalante T, Díaz C. Hemorrhage induced by snake venom metalloproteinases: biochemical and biophysical mechanisms involved in microvessel damage. Toxicon. 2005 Jun 15;45(8):997-1011. doi: 10.1016/j.toxicon.2005.02.029. Epub 2005 Apr 18. PMID: 15922771.
  3. Izidoro LFM, Sobrinho JC, Mendes MM, Costa TR, Grabner AN, Rodrigues VM, et al. Snake Venom L-Amino Acid Oxidases: Trends in Pharmacology and Biochemistry. BioMed Research International. 2014 Mar 12;2014:e196754.
  4. Kratzer E, Tian Y, Sarich N, Wu T, Meliton A, Leff A, et al. Oxidative stress contributes to lung injury and barrier dysfunction via microtubule destabilization. Am J Respir Cell Mol Biol. 2012 Nov;47(5):688–97.

Line 93 From how many specific individuals was the venom taken. How did they differ in age, sex, etc.?

Response: We inserted a new sentence in the methods Section explaining how venom was obtained as follows:

“The venom of Cdc was obtained from twenty adult snakes of both sexes kept at the Venomous Animals and Herpetology Laboratory of the Feira de Santana State University (LAPH-UEFS).”

Line 99. What method was used to measure the protein concentration in the venom, since the cited item (12) does not exist.

Response: We inserted a new sentence in the Methods Section explaining how we measured the protein concentration in the venom, which we present below: “Venom protein concentration was determined using bovine serum albumin (Sigma, Chemical Company) as the protein standard (Lowry, 1951).

LOWRY OH, ROSEBROUGH NJ, FARR AL, RANDALL RJ. Protein measurement with the Folin phenol reagent. J Biol Chem. 1951 Nov;193(1):265-75. PMID: 14907713.

Table 1, Figure 5a Please elaborate on the observed changes in the discussion. How is the decrease in inflammatory infiltrates at 6h after venom application and the number of inflammatory cells at 6 and 24h explained. Also, please note whether the commentary in the discussion in lines 299-306 is sure to describe the correct time points.

Response: We agree with the reviewer that more quantitative information about the modulation of pulmonary inflammatory injury would add relevance for the reader. We have revised the text, which we present below:

The increase of pulmonary inflammatory injury, mostly related to neutrophilic proliferation, especially at one, three, 24, and 48 hours, is partly explained by the release of macrophages chemotactic factors, epithelium disruption and proinflammatory effects of the venom related to crotoxin component phospholipase [42, 43]. However, inflammatory modulation with decreased neutrophilic infiltrates at six and 12 hours might be related to transient early neutrophil senescence and apoptosis induced by extensive free radicals’ production due to intense phagocyte activity [44-48].

Line 310 "Our data highlight the potential risk of early hypoxemic respiratory failure in the first hours after venom inoculation" Please indicate which specific results directly indicate this.

Response: We have revised the text in order to make our point clearer. We also provided a video of clinical signs of respiratory distress in experimental mice at supplementary material section.

“Our data showed early extensive necroinflammatory changes in lung parenchyma and clinical signs of respiratory distress, suggesting a potential risk of respiratory failure in the first hours after venom inoculation (supplementary material).”

Excerpt "Antivenom therapy is the specific treatment for snakebite envenomation and is essential to neutralize the venom toxins and prevent further tissue damage. Antivenom should be administered as early as possible, ideally within the first few hours after the snakebite" is unrelated in content to the earlier paragraph and has no citation. Please reword this passage.

Response: We agree with the reviewer that this sentence is not directly related to this paragraph. We have decided to suppress this sentence and include this topic in the last paragraph of the discussion, which we present below:

“Early clinical interventions during the first 6 hours of hospitalization guided by point-of-care biomarkers are critical to improve patient outcomes in high-risk subgroups and avoid long-term pulmonary sequelae [7,57]. Antivenom should be administered as early as possible, ideally within the first few hours after the snakebite [12]. Respiratory muscle paralysis and pulmonary edema remain the primary cause of death, with current treatment based on clinical and respiratory support [49]. Thus, experimental histo-pathological analyses can provide valuable insights for clinical trials in this field.”

My biggest complaint is with the list of cited literature. Items 10 through 14 do not exist. That is, there are articles with similar titles, but they are not published in the journals indicated by the authors. In the case of item 10, there is an article with this title in this journal, but written by other authors and almost 20 years earlier. In addition, in item 13 there are two citations, both non-existent. This makes half of the introduction written without support in the scientific literature. Moreover in item 41, the names of the authors are mixed up.

Response:  We agree with the reviewer that the introduction section must be improved. added new references to the introduction section and fixed reference 41.

Page 2 Line 51: “Crotalus durissus venom contains several enzymes and protein components that can induce biological effects, including myotoxins, phospholipase A2 (PLA2), L-amino acid oxidase (LAAO), phosphodiesterase (PDE), snake venom metalloproteases (SVMP), and serine proteases (SVSP) [10]. Crotoxin, constituted of PLA2 and crotapotin subunits, is one of the most studied toxins. PLA2 is an enzyme that hydrolyzes the phospholipids in cell membranes, leading to local and systemic myotoxicity, lymphatic injury, and edema [11]. In the respiratory system, PLA2-related neuromuscular paralysis of the bulbar and res-piratory muscles may be life-threatening [12].”

  1. Deshwal, A.; Phan, P.; Datta, J.; Kannan, R.; Thallapuranam, S.K. A Meta-Analysis of the Protein Components in Rattlesnake Venom. Toxins2021, 13, 372.
  2. Rangel-Santos A, Dos-Santos EC, Lopes-Ferreira M, Lima C, Cardoso DF, Mota I. A comparative study of biological activities of crotoxin and CB fraction of venoms from Crotalus durissus terrificus, Crotalus durissus cascavella and Crotalus durissus collilineatus. Toxicon. 2004 Jun 1;43(7):801–10.
  3. Gutiérrez JM, Calvete JJ, Habib AG, Harrison RA, Williams DJ, Warrell DA. Snakebite envenoming. Nat Rev Dis Primers. 2017 Sep 14;3(1):1–21.

Page 2 Line 60:  “SVMP hydrolyze structural components of the basement membrane, compromising the mechanical stability of pulmonary capillaries [12]. Therefore, Crotalus durissus venom can cause severe hemorrhagic disorders, inflammation, and pulmonary edema [13]. LAAO toxicity, primarily due to its enzymatic activity producing hydrogen peroxide and other reactive oxygen species (ROS), is related to local tissue damage and pulmonary injury [14]. Venom-LAAO catalyzes the oxidation of different groups of amino acids generating disturbances in blood coagulation and platelet aggregation. ROS are highly reactive molecules that can damage endothelial components and increase the expression of inflammatory genes. Excessive release of ROS causes oxidative stress and may lead to severe acute lung injury (ALI) [15].”

  1. Gutiérrez JM, Calvete JJ, Habib AG, Harrison RA, Williams DJ, Warrell DA. Snakebite envenoming. Nat Rev Dis Primers. 2017 Sep 14;3(1):1–21.
  2. Gutiérrez JM, Rucavado A, Escalante T, Díaz C. Hemorrhage induced by snake venom metalloproteinases: biochemical and biophysical mechanisms involved in microvessel damage. Toxicon. 2005 Jun 15;45(8):997-1011. doi: 10.1016/j.toxicon.2005.02.029. Epub 2005 Apr 18. PMID: 15922771.
  3. Izidoro LFM, Sobrinho JC, Mendes MM, Costa TR, Grabner AN, Rodrigues VM, et al. Snake Venom L-Amino Acid Oxidases: Trends in Pharmacology and Biochemistry. BioMed Research International. 2014 Mar 12;2014:e196754.
  4. Kratzer E, Tian Y, Sarich N, Wu T, Meliton A, Leff A, et al. Oxidative stress contributes to lung injury and barrier dys-function via microtubule destabilization. Am J Respir Cell Mol Biol. 2012 Nov;47(5):688–97.

We Revised Reference 41 accordingly.

  1. Hernández Cruz A, Garcia-Jimenez S, Zucatelli Mendonça R, Petricevich VL. Pro- and anti-inflammatory cytokines release in mice injected with Crotalus durissus terrificus venom. Mediators Inflamm. 2008;2008:874962.

Round 2

Reviewer 1 Report

I  understand the manuscript was a subsequent study of lung impairment induced by Crotalus durissus cascavella venom, the histopathological change was significant and was well manifested by the current work. However, the present work was not adequate enough for publication in Toxins. If possible, I hope the author to investigate the underlying mechanism of the impairment.

Reviewer 3 Report

Thank you to the authors for considering my comments. In my opinion, all the imperfections mentioned in the review have been corrected. In its current state, the article is suitable for publication. Please only at the proof-reading stage correct the fonts on pages 9 and 10.